# The Perceptions of Sexual Harassment among Adolescents of Four European Countries

**DOI:** 10.3390/children9101551

**Published:** 2022-10-13

**Authors:** Evanthia Sakellari, Mari Berglund, Elina Santala, Claudia Mariana Juliao Bacatum, Jose Edmundo Xavier Furtado Sousa, Heli Aarnio, Laura Kubiliutė, Christos Prapas, Areti Lagiou

**Affiliations:** 1Department of Public and Community Health, School of Public Health, University of West Attica, 11521 Athens, Greece; 2Laboratory of Hygiene and Epidemiology, School of Public Health, University of West Attica, 11521 Athens, Greece; 3Faculty of Health and Well-Being, Turku University of Applied Sciences, 20520 Turku, Finland; 4Nursing Research, Innovation and Development Centre of Lisbon, Nursing School of Lisbon, 1600-190 Lisbon, Portugal; 5Klaipeda City Public Health Bureau, LT-93200 Klaipeda, Lithuania

**Keywords:** sexual harassment, secondary schools, adolescents, prevention, health promotion

## Abstract

Sexual harassment is a crucial public health issue among adolescents. In order to develop school health promotion programs, there is a need to involve adolescents themselves paying particular attention to their perceptions, beliefs, attitudes and practices. Therefore, the aim of this study was to explore the adolescents’ perceptions about sexual harassment as well as the ways it could be prevented. Four focus groups were conducted during an online “camp” in autumn 2021, facilitated by members of SHEHAP project research team. Participants were secondary school students from Finland, Greece, Lithuania and Portugal. The qualitative data was analyzed using content analysis. Concerning how participants perceive sexual harassment, the themes that emerged were: physically expressed sexual harassment; verbally expressed sexual harassment; virtually expressed sexual harassment; violation of self-determination. Virtual environment; school environment; public environment; familiar environment, were identified as the places where sexual harassment may occur. Finally, in regard to the participants’ views on the prevention of sexual harassment, the following themes emerged: youth education; adult education aiming teachers and parents; professional, peer and family support; official consequences; health education methods. The findings of the current study can be used for the development of school-based programs aiming to prevent sexual harassment among adolescents.

## 1. Introduction

Sexual violence, including sexual harassment, frequently occurs in “safe” places, such as schools, where perpetrators include peers and teachers [1]. Studies have found differences in what perceived as sexual harassment according to characteristics of the behavior and context [2]. Sexual harassment is defined in the EU Directive (2006/54/EC) as “*where any form of unwanted verbal, non-verbal or physical conduct of a sexual nature occurs, with the purpose or effect of violating the dignity of a person, in particular when creating an intimidating, hostile, degrading, humiliating or offensive environment*” and it states that shall be accounted as discrimination on the grounds of sex and therefore prohibited [3].

A meta-analysis [4] has concluded that 1 in 5 adolescents have reported physical teen dating violence and about 1 in 10 reported sexual teen dating violence. In U.S.A., the Youth Risk Behavior Survey 2019 found that 8.2 percent of high school students reported physical dating violence and sexual dating violence, 10.8 percent reported sexual violence by anyone, with 50 percent of cases by a perpetrator other than a dating partner; 19.5% reported bullying in school, and 15.7 percent reported electronic bullying victimization during the past year [5]. In addition, a recent systematic review on teen dating violence among ten European countries concluded that there is a great variability in prevalence rates of psychological, physical, sexual and cyber teen dating violence victimization and perpetration with female adolescents is reported in higher rates in all forms of teen dating violence victimization than males [6]. Similarly, several studies report higher rates of sexual harassment among females [7,8] both online and offline [9] and it should be noticed that many women in Asia consider sexual harassment and violence a regular occurrence in their daily lives [10]. However, there was an earlier study in Israel which found higher reporting rates among males, explaining that this may be related to the fact that females interpret differently the violence in that society context while males are not ashamed to report that they were sexually harassed [11]. In addition, sexual orientation and gender identity are important factors, as LGBTQI youth seem to be particularly at risk for sexual harassment [12]. 

In regards to the different experiences of sexual harassment among young people, studies have found many different acts such as, showing offensive pictures or sending obscene letters, taking off or trying to take off part of the student’s clothing [13], verbal and physical sexual harassment [14] and places, suggest as, urban transport and other public spaces [13,15] or online in social media [14] with a recent study in a representative sample of US adolescents, reporting that nearly 15% of them having online sexual harassment experiences [16]. 

It was found that sexual harassment puts young people at risk for short-term and long-term health problems [17]. Negative experiences in the field of sexuality may be traumatizing during adolescence as it is the period when emotional development, the ability to cope with stressors and identity are still developed [18]. A recent study found that there are differences in depression scores between adolescents who reported online sexual harassment across their lifetime and participants who did not report it [19]. Another recent study concluded that online sexual harassment leads to increased anxiety and depressive symptoms among young females but not males [9]. Moreover, undesired sexual experiences in adolescence may increase later physical and sexual violence experiences, and the victim may commit in the future physical or sexual violence [20,21].

Although there are a number of studies among adolescents about its prevalence in specific groups, their experiences and the forms of sexual harassment, which is also described more in the literature presented above, there is not plethora of studies on the perceptions of adolescents about this essential public health issue [2,3,4,5,6,7,8,9,10,11,12,13,14,15,16,17,18,19,20,21,22,23,24]. Hence, the aim of this study was to explore the perceptions that adolescents have about sexual harassment as well as how it could be prevented. This study is part of the Erasmus+ project “Prevention of Sexual Harassment in Secondary Schools—SHEHAP” (2020-1-FI01-KA201-066493) whose aim is to develop digital health education material for secondary schools in order to prevent sexual harassment.

## 2. Materials and Methods

The study took place in October 2021 during a virtual “camp” as part of the learning/teaching/training activities of the SHEHAP project. Participants were pupils aged 13–16 years old. Data was collected using the focus group method. Focus groups have been chosen as a data collection method because of their unique advantage of participants interaction and they stimulate the discussion that may provide a wide range of attitudes, knowledge, and experiences [25]. Participants were divided in four transnational groups and facilitated by the researchers who have expertise in qualitative research and youth health. The online focus groups gave the opportunity to have mixed country groups which would enrich the discussions. Each focus group lasted about two hours. Notes were kept throughout the discussion in order to perform the content analysis later. During data collection anonymity was ensured. In the focus groups, the following same questions were asked: What is sexual harassment in pupil’s opinion? How and in what situations does sexual harassment appear in pupil’s daily life? How could sexual harassment be prevented at school?

Data was analyzed using content analysis and it was performed by five members of the research team (E.S., E.S., M.B., C.M.J.B.and J.E.X.F.S.) with expertise in qualitative research and youth health, and no software was used for the analysis. First, the researchers read the data several times in order to familiarize themselves with the data. Second, the researchers worked independently in order to identify the categories and subcategories. The final list of categories and sub-categories was developed after mutual consensus among the researchers. Figure 1 below presents an example of how categories were emerged (in the example below illustrates the different answers of the participants, the reader can find the original quotes and how these led to the categories).

Ethical approval to conduct the study was obtained from the Ethics Committee of Turku University of Applied Sciences (2021-066/31.5.2021). Because the participants were pupils under 18 years of age, their parents or guardians signed a written informed consent form. Information to the participants about the study, its process and objectives were provided. They were also informed about their right/ to withdraw from the study at any time they wish without giving any explanation and with no consequences. Participation was voluntary and no identifying information has been used to ensure the anonymity and confidentiality of the participants.

## 3. Results

The participants’, all pupils of secondary schools expressed their perceptions about sexual harassment and how it could be prevented providing rich information that is described below. The categories and subcategories that emerged from the data are presented below using examples of original quotations. Indeed, adolescents have been communicating their views and ideas on sexual harassment which were common among them even though they come from different ethnic and cultural backgrounds and their school environments may differ.

### 3.1. Perceptions of Sexual Harrassment

Concerning how participants perceive sexual harassment, the themes and subthemes that emerged are the following:Physically expressed sexual harassment, which included unwanted touching, physical violence, physical harassment. As participants exprssed their perception of sexual harrassment, it is:*…touching in an unwanted way…**…inappropriate touching…**…touching grabbing or making other physical contact with you without consent, unwanted touching…**…rape…violence…*Verbally expressed sexual harassment, expreseed in different ways as commenting, spreading rumours, joking, threating, blaming. As participants explained, sexual harrassment is:*…commenting in a sexual way about theirappearance…**…talking about someone’s private parts…**…slut shaming, making jokes, calling names…**…commenting about sexual orientation…**…making fun about sexuality…**…making sexual gestures and telling jokes or sexual comments about someone else…**…making someone feel bad if you don’t giving consent…*Virtually expressed sexual harassment in terms of sending harmful material or texting. For example, as participants said:*…sharing sexuality inappropriate images or videos…**…someone sending me their nude photos without asking me…*Violation of self-determination. As participants explained, this is about the actions without consent or unwanted sexual attention. For example, participants said:*…for us sexual harassment is disrespecting people, not caring about their premission, or their position in the subjet, but in the case the subject can be their body, their sexuality, or anything like that…**…without your permission…**…unwanted sexual attention…**…staring, looking at someone in an insulting, uncomfortable way…**…share someone’s intimate information or photos without consent…*

### 3.2. Environments That Sexual Harassment May Take Place

The following environments where claimed by the participants that sexual harassment may take place which include the environments of their everyday lives and not any exceptional places or circumstances.
Virtual environment. As particpants said in social media and the internet. As one participant said, for example:*…it happens…on social media…the internet…*School environment. As it was xeplained by the participants, it may take place in the school private places or there might be comments at school. For example as one of the participants described:*…at school—someone might make inappropriate comments about the growing of the body…*Public environment. Participants included several situations; at work, at hobbies, at public places, by strangers. As participants said:*…work life, between**colleagues…**…in free time, when doing sports, swimming, you might be swimming and someone makes a comment about your body…**…in streets…in public places, someone might give comments about your appearance…**…by strangers…a stranger follows you and gives you some comments…*Familiar environment. As participants described, sexual harrassment may occur at home and among friends as well. They said:*…at home between the parents, for example mom being harassed by dad…**…at a friend’s house, a friend makes accomment about you…*

### 3.3. Views on the Prevention of Sexual Harassment

In regards to the participants’ views on the prevention of sexual harassment, the following themes and subthemes emerged:Youth education which includes awareness, health education and open discussion. As participants said:*…educate students what is sexual harassment and what is being friendly, to recognize and distinguish those two…**…educate where you can find the help…**…debate the subject at school, so there is open discussion with everyone…*Adult education which should be addressed to teachers and parents as well. As supported by the participants:*…educate teachers about protecting kids from sexual harassment…**…teaching parents and teachers…*Support. This includes support received by a professional, a peer or the family. The participants*…have a professional, such as a psychologist, so that it is easier for the victim to speak about the situation…**…victims should see other victims and talk about and share their experience in order to support each other…**…parents should make a comfortable atmosphere for the children…**…need to have a very protective environment, so they won’t have a problem to discuss about this with their parents…*Official consequences. The participants claimed that immediate action should be taken and discussion is important to address the issue of sexual harrassment. They said, for example:*…to address the issue at the moment and not to ignore it…**…bystanders must seek help from teachers…**…we should face it, address it, openly discuss it and stop it at the moment it happens…*Health education methods. The participants provided their ideas about the methods that could be used in health education on sexual harrasssment prevention. These were, digital methods, books, movies, comics.

## 4. Discussion

This study explored the perceptions of sexual harassment among adolescents and their views on how to prevent this public health issue. The findings of this study indicate that adolescents from four European countries share similar perceptions and ideas about the sexual harassment issue providing a rich insight as presented above.

Concerning how participants perceive sexual harassment, the themes and subthemes that emerged were physically expressed sexual harassment; verbally expressed sexual harassment; virtually expressed sexual harassment and the violation of self-determination. Similarly, physically expressed sexual harassment was recognized in a study among female college students in Egypt, where the main perceived concept of sexual harassment was touching body (63.9 percent) followed by uncomfortable behaviors by the assault (51.8 percent) [26]. In previous studies, adolescents have described verbally expressed sexual harassment to appear, for example, as sexual jokes, sexual or demeaning comments, attractiveness rating, comments of clothing or as unpleasant insinuations. Verbally expressed sexual harassment seems to be more common than invasive, physically expressed harassment [2,27,28,29,30]. The literature has also identified online sexual harassment, for example, in a study among female adolescents in Croatia, the lifetime prevalence of online sexual harassment was 43 percent [31]. Another study among sexually active adolescent females, the majority (68 percent) reported at least one form of cyber sexual harassment, including receiving unwanted sexual messages or photos (53 percent), receiving unwanted messages asking them to do something sexual (49 percent), being pressured to send sexual photos (36 percent) and having sexual photos shared without permission (6 percent) [32]. In addition, a European study found that 15 percent of 11–16-year-olds had received sexual messages or images of people naked or having sex from their peers during the last year that the study was conducted [33]. Interestingly, earlier studies found that peer harassment is perceived differently by boys and girls, when boys are less likely to be frightened and they may perceive some potential harassment experiences as positive, while girls experience sexual harassment more often and experience qualitatively more severe, physically intrusive, and intimidating forms of harassment than the boys [34,35]. Sexting is also a confusing practice and youth perceptions about sexting definition and harmful content were different from those of most parents and teachers [36]. Finally, the self-determination concept was also present in the participants’ perceptions of sexual harassment in another study among adolescent girls and boys, adult women and men in Tanzania, who emphasized the critical role of consent [37].

The following settings were claimed by the participants to be more common for sexual harassment occurrence: virtual environment; school environment; public environment; familiar environment. Studies have shown similar environments that sexual harassment takes place. Similarly, a recent study in Nepal [38] concluded that school, roads and public places were the most common settings where participants encountered sexual harassment. In addition, college and school students, in Delhi, consider the public transport and bus stops as unsafe for them [13]. In regards to the virtual environment that sexual harassment may take place, it is also found that online sexual abuse among children, involves control, permanence, blackmail, re-victimization and self-blame [39]. It is important also to notice that young people are not able to identify the gender and age of the harasser, which shows the complexity of the phenomenon of online abuse [19]. It has been found that sexual harassment in the workplace is the most popular research theme, and sexual harassment has been investigated by several researchers in a wide range of places from school to military settings [40]. In a study in Ethiopia, among female pupils, it was found that sexual harassment mostly occurred in the schools, at 49.3 percent, followed by in classroom with 41.3 percent, and in the office at 19.7 percent [41]. Similarly, nursing students may experience sexual harassment during their clinical practicum, as a study reports more than one-fifth of them [42]. In an earlier study in Sweden, focusing on school environment, it was found that almost half of the female high school students who participated in the study (49 percent) identified sexual harassment as a problem in school and 15 percent believed this problem was serious [30]. Another study in South Africa found that both male and female university students had experienced sexual harassment on campus and female students were significantly more likely to be raped than male students [43]. In addition, a study about the experiences of Czech, Greek, and Norwegian female sport students found that 37 percent of them had experienced one or more forms of sexual harassment from someone in sport [44].

Finally, in regard to the participants’ views on the prevention of sexual harassment, participants supported the following: youth education; adult education, for teachers and for parents; professional, peer and family support; official consequences, with immediate actions to be taken. The literature supports the current study participants’ statements. It is supported also by the literature that preventive interventions lead to a reduction in assaults, related to sexual harassment. Implementing interventional strategies and education in adolescents can increase their awareness and attitude and moreover, reduce the consequences of assaults [45]. More specifically, it is supported that educating adolescents on social skills and appropriate ways of showing their interest in their peers, in order to not use sexual harassment as a means of expressing romantic interest, can all be beneficial for the prevention of this issue [46]. The results of a sexual assault prevention program showed a significant change in attitudes and it reduced incidences of assaults [47]. Another intervention which consisted of a classroom-based girls’ empowerment program and a boys’ educational program on gender equality conducted that the incidence of sexual assaults was significantly reduced in the group of students who attended the program [48]. It is also supported that parents may not be aware of the signs and symptoms associated with sexual abuse, while the more knowledge parents have, they can create safer environments for their children and therefore prevent the occurrence of sexual exploitation and hence it is found that parents can benefit from even brief educational efforts [49]. Teachers’ abilities to observe the occurrence of sexual harassment in the school environment may be deficient. On the other hand, even if the teacher suspects that sexual harassment is taking place, he may feel that he lacks the skills to raise concerns. Positive results have been achieved by training teachers by the matter, enhancing open discussion between adolescents and teachers and involving teachers in drafting rules and programs against sexual harassment [50,51]. In addition, as it is found in an earlier study, absence of punishment ranked first (54.1 percent) as one of reasons for the phenomenon of harassments [25], the participants of the current study support that official consequences can play a role in prevention. It is also supported by the literature that raising awareness to recognize and respond to different forms of harassment can be included in the strategies for tackling the problem [15]. In addition, the role of health professionals in prevention of sexual harassment and supporting adolescents and their social environment is essential by raising their awareness and improving their attitudes [52]. Their education allows them to identify risk factors among parents and adolescents and empower them [53]. Finally, support of victims/survivors are valuable in order to lessen harms, reducing short- and long-term negative effect and the risk for later perpetration among those victimized [54].

## 5. Limitations

A limitation of the current study is the use of a convenience sample which does not allow findings’ generalization, which is also due to the small number of participants based on the qualitative design approach. The fact that the participants had to express personal opinions in front of others may have influenced their responses. However, the focus groups have been multinational which allowed the expression of all views, there has been an open and friendly environment during the “camp” and also it can be seen that there has been saturation in the data collected.

## 6. Conclusions

It is suggested that further research should explore in more detail the role of the digital mediums in sexual harassment since young people use the social media platforms to socialize, connect with others and share material instantly.

This study provides an understanding on the adolescents’ views which should be taken into consideration for the development of school-based programs on sexual health education and particularly the prevention of sexual harassment. The findings of the current study show that the adolescents are very much aware of what can be sexual harassment and they also provide different aspects of where that occurs and how it can be prevented. Therefore, school sexual health education programs should provide the tools and develop proper skills to prevent and address the issue. Moreover, as the participants of the current study support, multi sectorial and participatory approach seem to be more effective in the prevention programs of sexual harassment episodes in school population; the views for the sexual harassment prevention include a holistic approach of health promotion in schools where a whole-school approach targets not only the pupils and teachers but also parents and therefore, create environments that promote health for all.

## Figures and Tables

**Figure 1 children-09-01551-f001:**
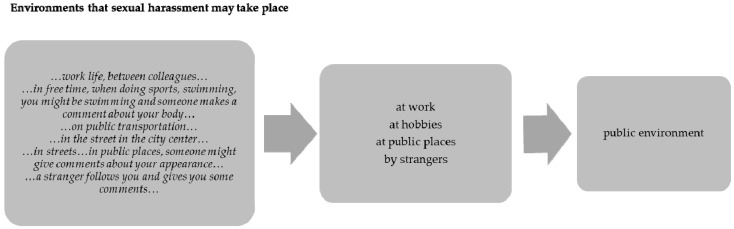
Example of how categories were emerged.

## Data Availability

The data are not publicly available due to ethical restrictions.

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
