# Peer review of "The Perceptions of Sexual Harassment among Adolescents of Four European Countries"

_children, 2022, doi:10.3390/children9101551_

Round 1
Reviewer 1 Report
Show a brief summary of the figures after showing the figure.
Line No.
49 - Write past tense, "It was found" ...
55- plural: there are ...
197- singular, "seems" instead of seem.
200- write percent when using within text. Use symbol (%)inI charts and graphs.
216 - use all lower case for common nouns.
249 - It may be a good idea to use a subheading for the limitation(s).
258 -Scratch the word, "However" and lead with this or something similar..
Suggest using italics for the journal title, helps with clarity from the article titles.
Author Response
We would like to thank the reviewer for the valuable comments to improve our manuscript all of which have been addressed (highlighted in the manuscript), please see below.
Show a brief summary of the figures after showing the figure.
Authors’ response: A description, please see lines 106-108.
Line No.
49 - Write past tense, "It was found" ...
Authors’ response: It has been corrected, please see line 69.
55- plural: there are ...
Authors’ response: It has been corrected, please see line 79.
197- singular, "seems" instead of seem.
Authors’ response: It has been corrected, please see line 226.
200- write percent when using within text. Use symbol (%)inI charts and graphs.
Authors’ response: It has been corrected throughout the text.
216 - use all lower case for common nouns.
Authors’ response: It has been corrected, please see lines 248-249.
249 - It may be a good idea to use a subheading for the limitation(s).
Authors’ response: A heading has been added, please see line 309.
258 -Scratch the word, "However" and lead with this or something similar..
Authors’ response: It has been deleted as suggested, please line 321.
Suggest using italics for the journal title, helps with clarity from the article titles.
Authors’ response: Italics have been used for the journal titles in all references.
Reviewer 2 Report
Dear Sir/Mam
Please find bellow the requested review regarding the manuscript. The article contains a lot of useful information on the issue. The topic is very interesting and but use of sources is not appropriate. Although it has some useful information there are less references and the statements are not established. I suggest the authors to write more information with references.
The article contains a lot of useful information on the issue. It is quite clear what is already known about this topic and the research question is clearly outlined. The abstract is too brief and introduction section involves too many information. The research question is not justified clearly, given what is already known about the topic. The results are not discussed from multiple angles and conclusions answer the aims of the study partially. The conclusions are partially supported by references or results and the limitations of the study fatal and it is questionable if there are opportunities to inform future research. Positive: There are some strengths of the article that could have an impact in the field, such as the topic and its impact on the existed literature. The manuscript is approved publication only after major changes.
Author Response
We would like to thank the reviewer for the valuable comments to improve our manuscript, all of which have been addressed (highlighted in the manuscript).
The abstract has been written according to the authors’ instructions with 200 words.
More information with references has been added and justification, supported by the literature has been added in introduction section, please see highlighted text, pages 1-2. It can be seen that 18 extra citations/references have been added.
More aspects and literature have been added in discussion in order to address reviewer’s comments (please see highlighted text, pages 6-11.
Limitations have been revised, please see limitations section (page 8).
Conclusions section has been revised according to the reviewer’s suggestions, please see conclusion section (page 8).
Reviewer 3 Report
Dear authors,
I congratulate you on the manuscript that deals with a very important social topic in a scientific and research way. Taking into account the importance of the topic, and the number of countries from which the participants of your focus group discussions come, I do not understand why the manuscript extends to only 9 pages. All parts of the manuscript should be deepened: Introduction - refer in more detail to the secondary literature on sexual harassment;
Method - describe the methodological framework of the research in more detail. What is the advantage of focus group discussions in such research, compared to other research techniques?
Results - this is the part of the work that requires the most intense "deepening".
Discussion - if you deepen the results, you will have to deepen the discussion.
State the limitations of the research
Author Response
We would like to thank the reviewer for the valuable comments to improve our manuscript, all of which have been addressed (highlighted in the manuscript).
The manuscript has been extended and more references have been added (please see highlighted text in the document).
In the methods, the choice of focus groups as a data collection method is justified (reference is provided), please see section “methods”.
Results have been revised in order for the readers to obtain a better understanding about the findings. Indeed, adolescents, as normally, communicate as usual, in short without providing long sentences always. In any case, examples of quotations are included in the results section in order for the reader to have a better overview. Please section “results”.
Discussion has been totally revised and more literature have been added in discussion in order to address reviewer’s comments (please see highlighted text, pages 6-11).
Limitations have been revised according to reviewer’s suggestion, please see section “limitations”, lines 319-323.
Round 2
Reviewer 2 Report
I have nothing further to add. I agree
Reviewer 3 Report
Thank you. The manuscript has been improved.